# An Inventory of South African Medicinal Plants Used in the Management of Sexually Transmitted and Related Opportunistic Infections: An Appraisal and Some Scientific Evidence (1990–2020)

**DOI:** 10.3390/plants11233241

**Published:** 2022-11-25

**Authors:** Nkoana I. Mongalo, Maropeng V. Raletsena

**Affiliations:** The College of Agriculture and Environmental Sciences (CAES), University of South Africa, Johannesburg 2092, South Africa

**Keywords:** South Africa, ethnobotany, survey, sexually transmitted infections, medicinal plants, antimicrobial activity

## Abstract

The current work is aimed at generating the first inventory of South African medicinal plants used in the treatment of sexually transmitted and related opportunistic infections associated with HIV-AIDS. This is important in assisting researchers to access a list of plant species to evaluate for potential phytocompounds, as this area of research is greatly lagging in South Africa. A total of 335 medicinal plants from 103 families have been documented in the current work. The most represented families are Fabaceae (11.64%) and Asteraceae (6.27%). Herbs constitute 36.53%, trees 32.34%, shrubs 29.04%, climbers 1.80% and parasites 0.30%. It is worrying that on the plant parts used, the roots constitute 47.18%, while leaves and stem bark yield 16.62 and 15.01%, respectively. *Catharanthus roseus* exhibited the highest number of citations (19), while *Peltophorum africanum* had 14 and both *Carica papaya* and *Vachelia karoo* had 12. In the mode of administration of the reported medicinal plant species, most of the plants are boiled and taken orally (48.22%), while other plant species are used as mouth washes (3.25%). Although there is reasonable in vitro activity of some of the plant species, validating the relevance of use, there is still a need to explore the mode of action of such plant species; isolated compounds and possible derivatives thereof are of paramount importance and need to be explored as well. Furthermore, toxicological aspects of such plant species need to be explored.

## 1. Introduction

Sexually transmitted infections (STIs) or venereal diseases (VD) are infections which are transmitted sexually from one partner to the other, through semen and/or blood. Such infections are treated with conventional drugs, which includes metronidazole, ciprofloxacin, ceftriaxone azithromycin and doxycycline [1]. Worldwide, there is an alarming increase in infections and much lower rate of discovering new drugs [2], particularly those used in the management and treatment of STIs and other illnesses considered critical by World Health Organization [3,4]. The development of microbial resistance to readily available antibiotics, particularly in developing countries, poses an enormous threat to both human and animal life. The development of many opportunistic infections associated with HIV-AIDS further results in many illnesses which may or not be easily treated with conventional drugs. According to Tshikalange et al. [5], although patients infected with STIs receive some antibiotics to treat infections, such patients still consult traditional healers or herbalists to purge the disease.

In recent times, researchers in Africa have developed an enormous interest in documenting medicinal plants traditionally used in many communities to manage and treat a variety of sexually transmitted infections [6,7,8,9,10,11,12,13,14,15,16,17]. It is important to note that such medicinal plants are readily available throughout the year, are socially acceptable to society and are believed not to possess any possible side effects [18,19]. In the current review work, the documented medicinal plants from various literature search engines, irrespective of origin, traditionally used in the treatment of sexually transmitted and related and resulting opportunistic infections associated with HIV-AIDS are reviewed. These exclude the medicinal plants reported in the treatment of tuberculosis, although they may be opportunistic to patients with HIV-AIDS. Furthermore, plant species used in the treatment of cancer and skin infections that may concurrently cohabit HIV-AIDS patients have also been omitted. The current work serves as a starting point to document possible alternative medicines and antibiotics that may well treat various STIs and opportunistic infections in South Africa. 

### 1.1. Sexually Transmitted Infections: Historical Context and Treatments Used

#### 1.1.1. Western Context (Focus on Gonorrhoea)

Sexually transmitted infections are infections that are acquired mainly through sexual contact and involve exchange of fluids that may enhance reproduction of many species. Such infections are known to be caused by many bacteria, parasites, and fungi such as *Neisseria gonorrhoea*, *Mycoplasma hominis*, *Mycoplasma genitallium*, *Chlamydia trachomatis* and *Klebsiella granulomatis* and some important viruses such as Cytomegalovirus, HSV type-2, HSV-8, Hepatitis B and human immunodeficiency virus [20,21]. 

Gonorrhoea is one of the most known and prevalent STIs and has been present from ancient times. It is known for its unwanted discharge of a whitish semen and is commonly called “*Pleasures of Venus*”, “*Strangury*”, “*The Clap*”, “*Les Clapiers*”, “*Rabbit huts*” and “*An issue of seed*”, as described in the Book of Leviticus in Old Testament and named by the Greek physician Galen [22]. The clap was derived from French brothels, known as “Les Clappiers”, while “*The Clap*” refers to a clapping sensation experienced by the infected penis, the male reproductive organ, that exudes pus. This disease was believed to be incubated in most women’s bodies, and men were believed to be victims while women were a cause. In the 16th century, mercury was injected into patients suffering from gonorrhoea as a form of treatment, while the insertion of catheter into the urethra with the intention of flushing the infection with a lot of warm water (at 45 to 50 °C) was also practiced in the 18th century [22]. This water procedure was administered twice within a short space of time, for a maximum period of 3 days. In the 19th century, Indonesian pepper and balsam extracted from a South American tree known as Copaiba were also incorporated as possible sources of treatment. However, it was later believed that Mercurochrome, a derivative of fluorescein with bromine and mercury, was a better remedy compared to all other medicines used, particularly in the world wars when soldier was infected with the disease [22]. In 1913, a heat therapy was advocated as a cure for gonorrhoea, while silver nitrate was also employed. 

Penicillin is one of the first antibiotics discovered back in 1929 by Sir Alexander Fleming from moulds, and it is still the greatest milestone in modern medicine [23]. In 1950 and 1953, tetracycline and erythromycin were then introduced, respectively, as the most potent antimicrobial agents used against major human, and this regime was termed a golden era [2]. In 1987, a fluoroquinolone called ciprofloxacin was introduced as a broad-spectrum antimicrobial agent that treated both syphilis and gonorrhoea [24]. Most antibiotics used in developing countries in the treatment of STIs to date include penicillin, streptomycin, tetracycline, ciprofloxacin, azithromycin, metronidazole, clindamycin, and others. It is important to note that from 1940 to date, pathogenic microorganisms, particularly those implicated as causative agents of STIs and related opportunistic infections, are gradually developing resistance to many commonly used antibiotics [25]. Some authors propose the use of medicinal plant extracts and/or in combination with readily available antibiotics to enhance biological activity [26]. However, safety standards need to be considered.

#### 1.1.2. African Context

The use of plants in treating many pathogenic infections against variety of microorganisms is ancient in Africa. However, such ethnobotanical data are not well documented, and this remains so to date [27]. Furthermore, the scientific evidence and validation of such medicinal plants, particularly those used in the treatment of sexually transmitted infections, remains unknown to the lay people in Africa. Such traditional knowledge is believed to be part of the customs and values of various communities, passed on from one generation to the other and believed to be highly holistic, as it involves both the body and the mind [27]. Furthermore, such medicinal plants are known to possess several advantages such as fewer side effects, low cost and being readily available [28]. Such plants are used mostly in combination with other plants to alleviate antimicrobial resistance, which is common in Western methods of healing. Although not scientifically validated, such plants are also believed to purge the disease [5]. In South Africa, the use of plant species such as *Arctopus monacanthus* Carmich. ex Sond, synonym *Arctopus echinatus* L., in the treatment of common sexually transmitted infections such as gonorrhoea and syphilis by the early Cape dwellers dates to the Thunberg’s travels in the eighteenth century [29]. It is also important to note that the Indian labourers, from Tamil Nadu, later arrived in the Province of KwaZulu-Natal in South Africa in the 1860s to work in the sugarcane industry. Later in the 1870s, Gujarati-speaking Hindus and Muslims from the west coast of India were also introduced in South Africa and pioneered Indian trading in KZN, mainly to “serve the needs of the Indian labourers.” These two groups of Indians are believed to have started the use of plants from India in treating a variety of STIs [30,31] and then introduced Ayurvedic medicine and alien plants into South Africa.

#### 1.1.3. Relationship between Sexually Transmitted Infections and HIV-AIDS

Human immunodeficiency virus (HIV) remains the only incurable type of STI, which may result in several various infections within a human body and result in a syndrome. Such infections may result in many opportunistic infections, which may well be treated using both antibiotics and medicinal plants [32]. It is important to note that an STI infection always increases the chances of HIV infection, as both may be acquired through unprotected intimacy (primary transmission) and from a pregnant mother to the unborn baby (secondary transmission). According to Rebbapragada and Kaul [33], STIs enhance both HIV susceptibility and secondary transmission. Furthermore, mechanisms that underpin this negative synergy between a specific STI and HIV include the impairment of innate mucosal defences, induction of pro-inflammatory cytokines, origin of activated immune cells that might well enhance HIV replication, increased susceptibility to other STIs and alterations in the normal microflora in the vagina. HIV, which may be either HIV-1 or HIV-2, infects CD4+ T-lymphocytes and results in CD4 cells and a lowered immune system, which allows multiple bacterial and fungal infections [34]. Such infections are called opportunistic infections and include raised lesions on the skin on the whole body, hairy leucoplakia, vaginal ulcers (vaginosis), lesions on the tongue and or mouth (oral candidiasis) and regeneration of tuberculosis or other complicated respiratory infections [35,36]. In African traditional medicine, patients are usually administered a decoction which is believed to improve immunity, thereby alleviating many of these opportunistic infections.

Since both the approaches in ancient times and the current Western methods of healing had some shortfalls, which may include the antimicrobial resistance of the microbes implicated as causative agents of sexually transmitted infections, the use of plant species is one of the solutions. Such plants are readily available, natural, socially acceptable to society and may well serve as the first line of defence against devastating human pathogens [5,7,9].

## 2. Results and Discussions

Data on medicinal plant species used in the treatment of sexually transmitted and related opportunistic infections, particularly associated with HIV-AIDS, were scrutinized using several search engines. A total of 63 studies were used to document the medicinal plants used in the treatment and management of sexually transmitted and related opportunistic infections. Such literature studies (Table 1) and indicated sources originate from a total of 47 published papers, 12 theses and 4 books. Although some literature sources were not identified (9) by location, hence perceived as countrywide, reported sources were distributed into six provinces, Limpopo (38), Mpumalanga (5), Western Cape (1), KwaZulu-Natal (6) and Eastern Cape (4) as shown in Figure 1 below. Amazingly, there were no report found in the three provinces Gauteng, North-West Cape and Northern Cape. These results corroborate the study by Mongalo and Makhafola [19], which referred to many parts of Limpopo Province as a hotspot for medicinal plants with rich plant diversity. However, it is important to note that most of the surveys might have yielded fewer plant species due to the reluctance of informants to offer information and/or the lack of researchers capable of extracting information from various communities [10]. Language barriers, lack of knowledge and interest of scientific names of plants, stigma associated with HIV-AIDS and other factors might serve as setbacks in documenting such medicinal plants [37].

### 2.1. Diversity of Plant Species Used to Treat Sexually Transmitted and Related Infections

In the current work, a total of 335 medicinal plants from 103 families Appendix A are reported, and the most represented families include Fabaceae (11.64%), Asteraceae (6.27%), Apocynaceae (3.88%), Solanaceae (3.88%), Anarcadiaceae (2.99%) and Euphorbiaceae (2.99%). Only well-represented families with five or more species reported are shown (Figure 2). Elsewhere, from other countries, species from Fabaceae family have been reported to be prevalently used in the management of STIs and related opportunistic infections associated with HIV-AIDS [15,17], corroborating the findings in our current work. In Africa, Fabaceae, Asteraceae and Rubiaceae are considered as the top three dominant families in African medicinal flora [96]. The Fabaceae family comprises a variety of ethnopharmacologically important herbs, shrubs, trees and vines, yielding about 600 genera with 12,000 species [97]. Phytochemically, the family comprises a wide variety of compounds that includes contains important metabolites such as alkaloids, anthraquinones, flavonoids, tannins, glycosides, steroids, terpenoids, saponins and volatile oils [98]. The prevalence of these various compounds may well explain the ethnomedicinal usage and pharmacological relevance of the species. 

Out of 335 medicinal plants reported in the current work, herbs constitute 36.53% (Figure 3), followed by trees (32.34%), shrubs (29.04%), climbers (1.80%) and parasites (0.30%). 

A similar trend has been reported in South African medicinal plant species used in mitigating pain and inflammation, which involves illnesses such as toothache, general pain, headache, swelling, back ache, stomachache, menstrual or period pains and the disturbance of normal functions of different body parts [99]. Klopper et al [100] described South African plant biodiversity as “richest of the rich”, yielding a higher diversity of plants in the Cape Floristic Kingdom, the richest of the world’s hotspots of plant diversity, the Succulent Karoo biome, the most speciose arid zone in the world, and the Maputaland–Pondoland–Albany Region. Furthermore, most of such plant species are used medicinally for multiple disease management, either individually or in combination with others [5,19,66,69].

### 2.2. Citations of Plants Used to Treat STIs and Related Infections

A total of 29 medicinal plant species exhibited a number of citations ≥6 (Figure 4), with *Catharanthus roseus* (L.) G.Don (19) yielding the highest number of citations followed by *Peltophorum africanum* Sond. (14), *Carica papaya* L. (12), *Elephantorrhiza burkei* Benth *Hypoxis haemerocallidea* Fisch., C.A.Mey. and Avé-Lall., *Senna petersiana* (Bolle) Lock, *Ximenia caffra* Sond. (11) and both *Solanum panduriforme* E.Mey. and *Securidaca longipedunculata* Fresen. Yielding 10 citations apiece. These plant species have also been reported in the treatment and management of many other human and animal infections. *C. roseus* is used in the management of diabetes, various types of cancers, hypertension and malaria, while *P. africanum* is recommended in the treatment and management of illnesses such as bilharzia, eye infections, joint and back pain, toothache, ascites and abdominal disorders, diarrhoea, dysentery, infertility, skin rashes and blisters and depression [101,102,103].

According to Tlakula [85], and Rankoana [72], the use of medicinal plants in treating infections is heavily reliant on indigenous knowledge of individuals and the availability of such medicinal plant species. In African traditional medicine, the use of medicinal plants in treating infections is generally holistic in nature, hence for the treatment of many infections, blood-cleansing plant species are incorporated into the plant medicines with the intention of purging the disease [104]. Such plant species are generally red in colour and highly likely to include *Elephantorrhiza* species [19]. The treatment of STIs and related opportunistic infections require the general removal of toxins and detoxification of the body. The syndrome experienced by most HIV-AIDS patients results in deterioration of the overall immune system. This is further compounded by the resistance exhibited by many microorganisms that cohabit the host [105]. Medicinal plants remain an alternative source of medicine to treat and manage such infections [55].

### 2.3. Mode of Administration and Plant Parts Used to Treat STIs and Related Infections

Out of the 335 medicinal plant species reported in the current inventory, roots constitute 47.18%, followed by leaves (16.62%), stem bark (15.01%), whole plant (10.46%), bulbs (6.43%), fruits (4.02%) and then twigs at 0.27% (Figure 5). The prevalent use of roots as a source of medicine in treating other devastating human illnesses has been reported elsewhere [106]. The use of whole roots and root bark (underground plant materials) is extremely detrimental to general plant life, hence is highly likely to result in death [19]. These is likely to reduce both medicinal plant diversity and richness alike. The use of leaves is of paramount importance, as it is a possible conservation measure [107]. It is important to lead the research on the use of the leaves from similar medicinal plants as those traditional healers and herbalist recommend for the treatment of STIs and related infections. These may assist in conserving the medicinal plant diversity and richness. 

Preparation and mode of administration of medicinal plants reported to manage STIs and related infections are reported in Figure 6. This is dominated by the medicinal plants which are boiled and taken orally (48.22%), followed by not mentioned (35.18%), infusions taken orally (4.73%), mouth wash (3.25%), sap or powders applied directly to the skin (3.25%), powders licked by tongue (2.37), inhalation and steaming (1.18%), enema (0.89%) and squeezed (0.30%). The preferred method of preparation and administration of plants is using boiled medicinal plants (decoctions) taken orally. A similar trend has been reported elsewhere [108]. In these instances, the medicinal plants are either boiled individually or in combination [43]. The combination of such medicinal plants is believed to increase the biological activity of such decoctions, yielding a synergistic effect [66].

### 2.4. Relevance and Some Scientific Evidence of South African Medicinal Plants Used to Treat STIs and Related Infections 

#### 2.4.1. In Vitro Antimicrobial Activity

The biological activity of some of the reported medicinal plants are reported (Table 2). The in vitro biological activity against a plethora of microbes implicated as causative agents of STIs and related infections was explored. The various extracts and isolated compounds exhibited varying degrees of antimicrobial activity against pathogenic microbes. Although there is no standard on the potent antimicrobial activity of plant extracts, plant-based products and the isolated compounds, the consensus is that a potent antimicrobial plant extract or fraction should exhibit an MIC value of 1 mg/mL or less [109]. Contrarily, other authors recommend the extracts and isolated compounds to be highly active against pathogens at 0.1 mg/mL [110]. In the current work, the fractions with MIC values of 0.1 mg/mL or less are referred to as highly potent [111]. 

Furthermore, fractions with MIC values ranging from 0.1 to 0.3 mg/mL possess moderate antimicrobial activity, while fractions with an MIC value >0.3 mg/mL are referred to as inactive. Judging by these standards, the extracts from *Helichrysum caespititium, Peltophorum africanum*, *Carica papaya*, *Grewia flava* and *Aloe ferox* are amongst the plant species which yielded a noteworthy and potential antimicrobial activity against agents of STIs and related opportunistic infections. However, the mode of action of such medicinal plant species still needs to be explored. Furthermore, there is a need to explore the antimicrobial activity of the medicinal plant species reported in the current inventory Appendix A, which needs to be evaluated against microbes implicated as causative agents of sexually transmitted and related infections. Although the traditional and laypeople of South Africa prefer using underground plant materials and stem bark, there is a need to consider using the leaves for conservation purposes.

#### 2.4.2. In Vitro Antiviral Activity

HIV-AIDS continues to pose an unprecedented public health problem of enormous proportions to society worldwide, with no cure. Current treatment options for HIV-AIDS have not been satisfactory, and the quest for effective curative or preventive therapies goes on. Several countries have proposed the use of a wide combination of plant species [118]. However, the herb–drug interaction in such patients may well result in many complications that may include diarrhoea, vomiting and other devastating health effects [119]. According to Prinsloo et al. [120], there is an enormous interest in the use of medicinal plants in treating and managing HIV-AIDS in Africa. Water, butanol and ethyl acetate fractions from *Peltophorum afracanum* butanol extract exhibited a noteworthy anti-HIV-1 activity, yielding IC_50_ (50% concentration that inhibits the viral growth) values of 1, 3 and 4 µg/mL, respectively, in a MAGI CCR5 assay [121]. Furthermore, betulinic acid isolated from the butanol fraction exhibited IC_50_ values of 0.04 and 0.002 µg/mL against two different strains of HIV-1 and the highest selectivity index (SI). The higher the SI, the higher the safety margin [66]. It is important to note that betulinic acid is abundant in many South African medicinal plants with many other important biological activities [46,118,121]. Furthermore, the synthesis of the other compounds from betulinic acid yielded an increased biological activity [47,122]. Elsewhere, Bessong et al. [123] reported an IC^50^ value of 3.5 μg/mL against the RNA-dependent DNA polymerase (RDDP) activity of RT by the methanol extract of the stem bark of *Peltophorum africanum* Sond. Furthermore, the gallotannin isolated from the stem bark inhibited the RDDP and RNase H functions of RT with IC_50_ values of 6.0 and 5.0 μM, respectively. 

#### 2.4.3. Toxicity and Clinical Trials

The use of hypoxide isolated from *Hypoxis haemerocallidea* in a clinical trial for patients with cancer and HIV-AIDS resulted in anxiety, nausea, vomiting and diarrhoea and apparent bone marrow suppression [118,124]. Furthermore, the in vivo studies revealed that both extracts and hypoxide did not alter the pharmacokinetics of efavirenz or lopinavir/ritonavir [125]. Elsewhere, Lamprecht et al [126] studied the anti-HIV activity of some compounds from *H. haemerocallidea* in rats infected with HIV and found that the CD4 count of rats treated with phytosterols from the plant species had stable CD4 cell counts compared to placebo, a control drug, and that the mortality between the two groups was significantly different. In another study, 37 male tuberculosis patients treated with phytosterols in combination with TB drugs resulted in significant weight gain compared to those treated with drugs alone [127].

*Peltophorum africanum* was evaluated for its cytotoxic effect against a plethora of human cell lines [102,128]. The acetone extracts from roots, stem bark and leaves did not show toxicity in the brine shrimp and the Vero monkey kidney cell line assays, at a concentration of 5 mg/mL, each yielding LC_50_ > 1000 μg/mL [101]. Although Okeleye [129] reported 96% ethyl acetate extract to possess some degree of toxicity at concentrations ranging from 5 to 1000 μg/mL, especially after 72 h on a human Chang liver cell line, there is still a need to explore the cytotoxicity of the medicinal plants used in the treatment and management of sexually transmitted and related infections. It is important to note that most of such plant species are administered orally three times a day and are not dependent on the weight of the patient. 

In addition to exhibiting a potent enhanced cell migratory activity of endothelial hybrid and fibroblast cells in vitro, various extracts from *Terminalia sericea* were found to be not cytotoxic to a variety of cell lines, including SC-1 fibroblasts, the EA.Hy926 endothelial hybrid, and Vero cell lines [130,131]. Although clinical trials are of paramount importance in ethnopharmacology, it is important to note that South African research in that regard is lacking. However, there is a growing interest in *in vitro* toxicological studies [46,47,66,101,112,113]. 

### 2.5. Plant-Based Mixtures and Products Used in Treating Sexually Transmitted and Related Infections

There are a variety of plant-based mixtures sold over the counter used in the treatment of a variety of STIs and related opportunistic infections (Table 3). However, these mixtures have a plethora of unidentified plant names, quantities used and other important relevant pharmacological indications. It is important to note that these mixtures pose a tremendous danger to humans, as their toxicological and pharmacological aspects are overlooked [132]. Such plant-based mixtures reportedly exhibit some devastating mutagenic effects against the human reproductive system and may well lead to infertility and some complications that relate to impotence and infertility [133]. 

Elsewhere, a decoction made up of about nine medicinal plant species has been evaluated for anti-HIV-1 effects in vitro [70]. The decoction exhibited a noteworthy antiviral activity, yielding an IC_50_ value of 0.18 mg/mL, while Combivir and Kaletra yielded IC_50_ values 0.06 and 0.30 mg/mL, respectively. It is possible that the plant mixture comprised high levels of betulinic acid, as *Peltophorum africanum* was a major ingredient. It is important to note that these reported plant-based mixtures are not available over the counter. Furthermore, the quantities of such plant materials are not reported, thereby making it difficult to evaluate the biological activity in vitro. Although the mixture exhibited potential ant-HIV-1 activity, there is a need to explore the in vivo activity to further ascertain the anti-HIV activity. This section may be divided by subheadings. It should provide a concise and precise description of the experimental results, their interpretation, as well as the experimental conclusions that can be drawn.

## 3. Materials and Methods

### 3.1. Strategy for Literature Search

The information reported in the current paper was collected from a literature search using various computerized databases such as ScienceDirect, Scopus, Scielo, Scifinder, PubMed, Web of Science and Google Scholar. Additional information was further retrieved from various academic dissertations, theses and general plant sciences, ethnomedicine and other relevant ethnobotanical books. This was conducted following (covering from 1990 until December 2020) the guidelines by the Preferred Reporting Items for Systematic Reviews and Meta-Analyses (PRISMA) statement [134]. Key words such as South Africa, medicinal plants, sexually transmitted infections, opportunistic infections associated with HIV-AIDS, ethnomedicinal uses, survey and ethnopharmacological aspects were used interchangeably. The limitation of the current study is primarily the exclusion of the plant species traditionally used in the treatment and management of tuberculosis.

### 3.2. Data Mining

To generate the inventory/data Appendix A, the inclusion criteria were the following: (1) the literature has ethnobotanical or ethnopharmacological context, and articles should be ethnobotanical field studies/surveys reporting on plant(s) with an indication as used for treating sexually transmitted infections (STIs) and related conditions; (2) the study location must be South Africa; (3) study must focus on plants; (4) study must be written in English. On the other hand, the exclusion criteria were the following: (1) articles with no scientific plant names; (2) review articles; (3) articles focusing on animals and other natural resources used for treating STIs, and other conditions such as tuberculosis. Plant species reported for treatment of urinary tract infections, infertility, mouth ulceration without indication on HIV-AIDS, increasing libido, bladder complaints, cracked lips and purgatives were also excluded. Plant species were further verified using SANBI (South African National Biodiversity Institute) and The Plant List website (http://www.theplantlist.org, accessed on 9 October 2022). The data were collected with help from library staff at the University of South Africa (Florida Campus). In the search engine, plant species with only genus name were omitted in the current work. 

The papers, books and other sources used in the current work were screened for inclusion, ranging from 1990 to 2020, two decades. Papers appearing as duplicates, cited in abstract form or not in English were excluded (Figure 7). The task was conducted by the first author and confirmed by the second author. From each of the relevant articles, the scientific names, family, plant parts, method of preparation and use in the management and treatment of sexually transmitted and related opportunistic infections were recorded. 

## 4. Conclusions

There is enormous traditional knowledge on the use of medicinal plants in mitigating STIs and related infections in South Africa. In the current work, the first inventory of 335 medicinal plants was constructed and it is evident that their utilization for managing STIs and related infections remains a common practice in South African folk medicine. It is important to note that the recorded plants were utilized among the different ethnic groups. The most common species are from the family Fabaceae (11.64%). Herbs constitute 36.53%, followed by trees (32.34%), while *Catharanthus roseus* (19) and *Peltophorum africanum* (14) were the plant species with high citations from various literature sources. Several plants are used, in combination, to prepare medicines which are also available over the counter; however, the safety profiles of such medicines are still overlooked. Interestingly, these two species mentioned above further exhibited a potent antimicrobial activity against etiologic agents of sexually transmitted infections and some important microbes which are major causative agents of opportunistic infections such as *Candida albicans* and *Staphylococcus aureus*. A total of 29 recorded plants are relatively well known, given that they were mentioned by six or more sources in South Africa. In some cases, it was observed that important information such as preparation methods and recipes for a significant portion of identified plants were missing in the existing literature. This underscores the limited value of the existing fragmented nature of the ethnobotanical surveys in South Africa. To mitigate these challenges, adherence to the established guidelines and robust ethnobotanical research methodology remains essential for the development of a holistic inventory relating to remedies used for the treatment of STIs and related infections in South Africa. It is important to note that these medicinal plants may well serve as complementary and alternative medicine, particularly in remote rural areas without hospitals, to the available antibiotics which are under pressure due to antimicrobial resistance and the costs of various pharmaceutical products.

## Figures and Tables

**Figure 1 plants-11-03241-f001:**
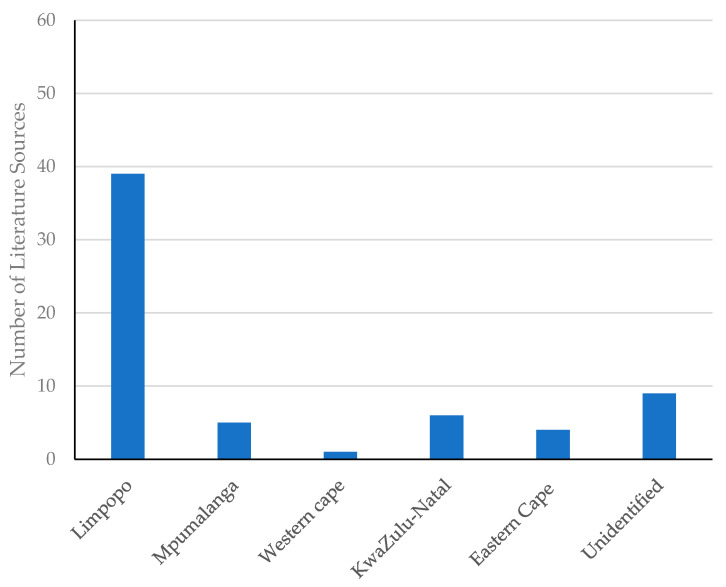
Distribution (by Province) of the literature sources used in the current study.

**Figure 2 plants-11-03241-f002:**
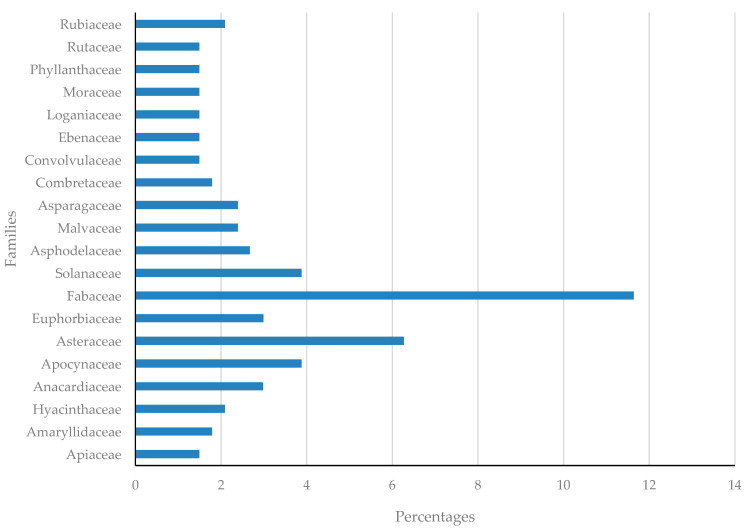
The most represented families in percentages (with at least 5 plant species or more).

**Figure 3 plants-11-03241-f003:**
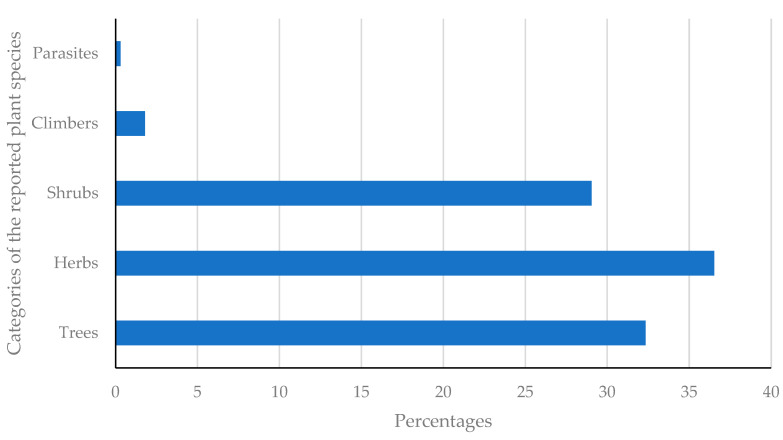
Percentages of the various categories of plants reported.

**Figure 4 plants-11-03241-f004:**
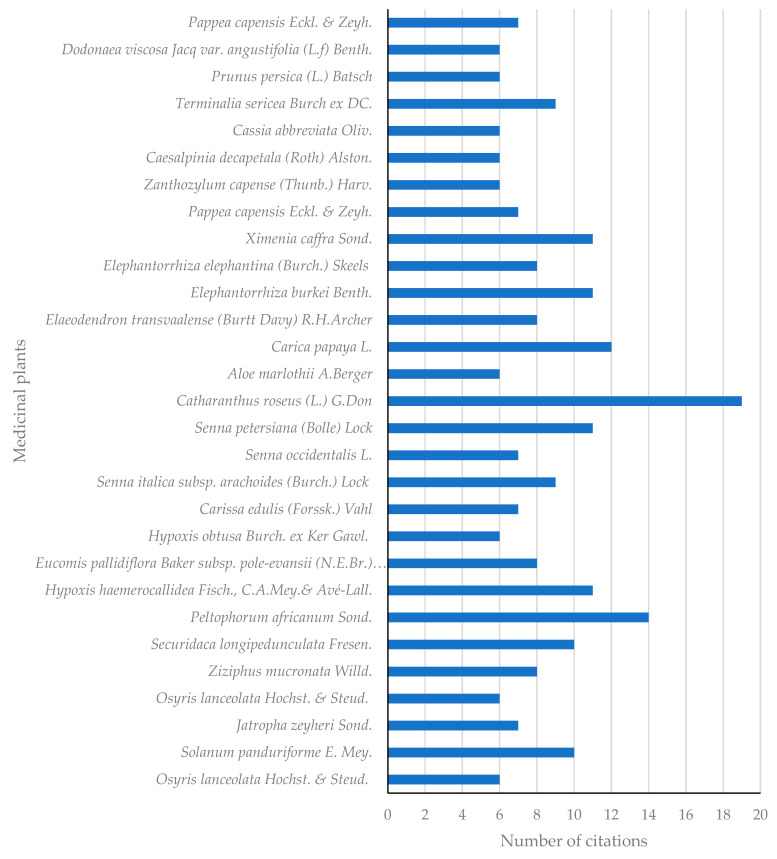
Plant species with higher number of citations (≥6).

**Figure 5 plants-11-03241-f005:**
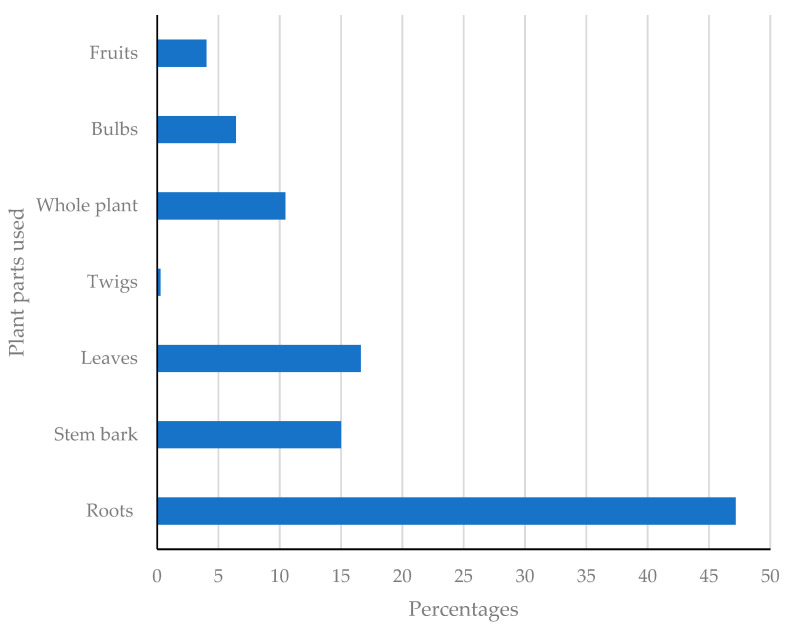
Various percentages of the plant parts used to treat sexually transmitted and related infections.

**Figure 6 plants-11-03241-f006:**
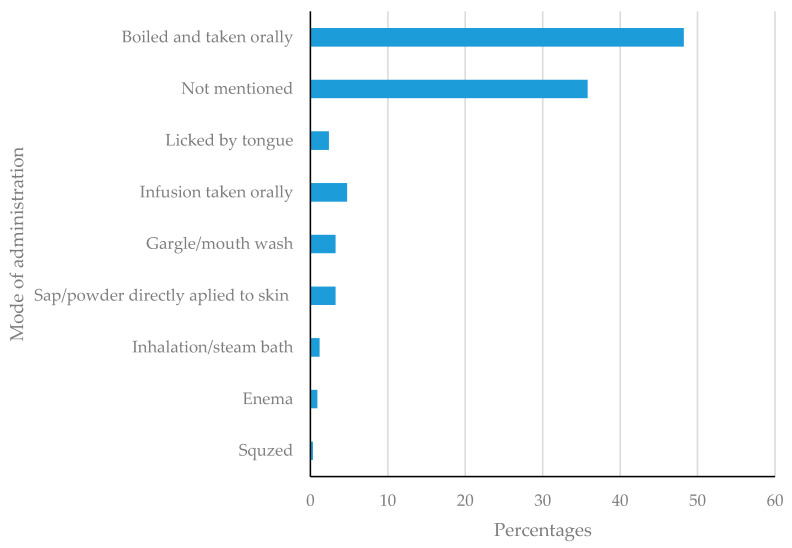
Preparation and mode of administration of medicinal plants reported to manage STIs and related infections.

**Figure 7 plants-11-03241-f007:**
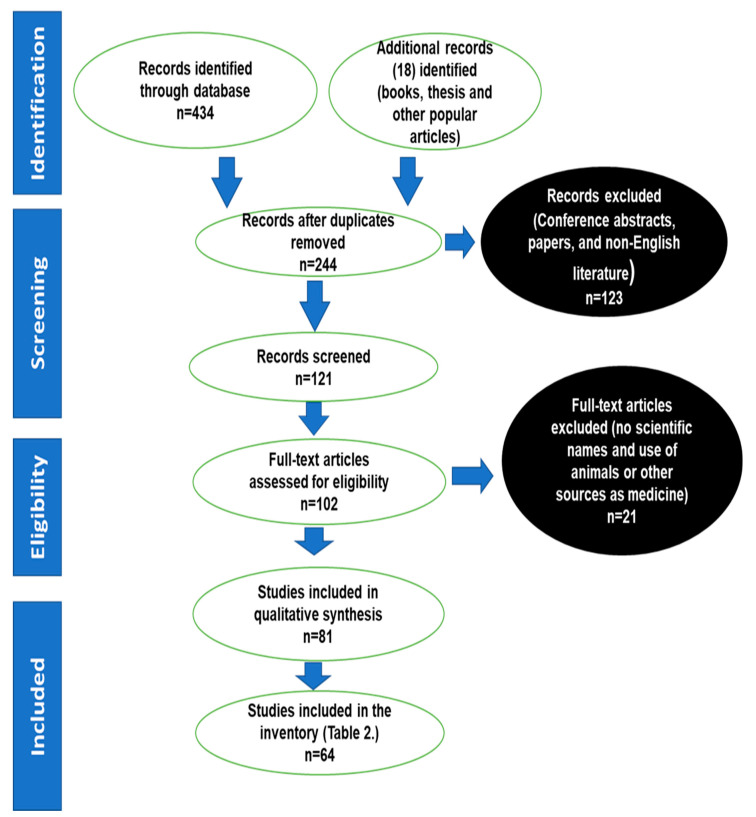
Sources extracted from the literature and used to collate the inventory of medicinal plants used to treat STIs and related opportunistic infections.

**Table 1 plants-11-03241-t001:** An overview of literature documenting the use of medicinal plants for managing and treating STIs and related opportunistic infections in South Africa.

Sources	Province	Title or Focus of the Study	Citations
[5]	Limpopo	Antimicrobial activity, toxicity, and isolation of a bioactive compound from plants used to treat sexually transmitted diseases.	6
[9]	Eastern Cape	An ethnobotanical survey of medicinal plants used by traditional health practitioners to manage HIV and its related opportunistic infections in Mpoza, Eastern Cape Province, South Africa.	17
[18]	Limpopo	The traditional use of medicinal plants to treat sexually transmitted diseases.	6
[19]	Limpopo	Ethnobotanical Knowledge of the Lay People of Blouberg Area (Pedi Tribe), Limpopo Province, South Africa.	33
[29]	Eastern Cape	Ethnobotany and antimicrobial activity of sieketroos.	3
[37]	KwaZulu-Natal	Zulu Medicinal Plants: An Inventory.	13
[38]	Eastern Cape	Antibacterial and antifungal activity of traditional medicinal plants used against venereal diseases in South Africa.	13
[39]	Eastern Cape	Biological activity of traditional medicinal plants used against venereal diseases in South Africa.	13
[40]	Limpopo	Medicinal plant use of villagers in the Mopani district, Limpopo province, South Africa.	21
[41]	KwaZulu-Natal	Ethnobotanical plant uses in the KwaNibela Peninsula, St Lucia, South Africa.	5
[42]	Unidentified	An ethnobotanical survey of the Agter-Hantam, Northern Cape Province, South Africa.	4
[43]	KwaZulu-Natal	Medicinal plants used for the treatment of sexually transmitted infections by lay people in Northern Maputaland, KwaZulu-Natal Province, South Africa.	33
[44]	Limpopo	Phytomedicine versus Gonorrhoea: The Bapedi experience.	18
[45]	Limpopo	Erectile dysfunction: Definition and materia medica of Bapedi traditional healers in Limpopo Province, South Africa.	12
[46]	Mpumalanga	In Vitro evaluation of the comprehensive antimicrobial and antioxidant properties of Curtisia dentata (Burm.f) C.A. Sm: Toxicological effect on the Human Embryonic Kidney (HEK293) and Human Hepatocellular carcinoma (HepG2) cell lines.	1
[47]	Mpumalanga	Isolation of anti-mycobacterial compounds from Curtisia dentata (Burm.f.) C.A. Sm Curtisiaceae.	1
[48]	Limpopo	In vitro activity medicinal plants of the Venda region, South Africa, against Trichomonas vaginalis.	29
[49]	Limpopo	The ethnobotany of the VhaVenda.	45
[50]	Limpopo	Ethnobotanical survey of invasive alien plant species used in the treatment of sexually transmitted infections in Waterberg District, South Africa.	14
[51]	Limpopo	Data set on preliminary phytochemical analysis and antioxidant activity of selected invasive alien plant species used in the treatment of sexually transmitted infections in Waterberg District, South Africa.	6
[52]	Limpopo	A quantitative survey of the traditional plant use of VhaVenda, Limpopo Province, South Africa.	55
[53]	Limpopo	An ethnobotanical survey of indigenous knowledge on medicinal plants used by the traditional healers of the Lwamondo area, Limpopo Province, South Africa.	5
[54]	Unidentified	Evaluation of the antimicrobial properties of Vachellia karroo Hayne Banfi and Galasso pods used traditionally for the treatment of venereal diseases.	1
[55]	Limpopo	Medicinal plants and traditional practices in peri-urban domestic gardens of the Limpopo Province, South Africa.	7
[56]	Limpopo	The traditional use of plants to manage candidiasis and related infections in Venda, South Africa.	45
[57]	Limpopo	Isolation and characterization of antifungal compounds from Clerodendron glabrum var glabrum (Verbenaceae) used traditionally to treat candidiasis in Venda, South Africa.	45
[58]	Limpopo	Medicinal plants used to manage sexually transmitted infections by Bapedi traditional health practitioners in the Blouberg area, South Africa.	17
[59]	Limpopo	Invasive alien plants used in the treatment of HIV-AIDS-related symptoms by traditional healers of Vhembe Municipality, Limpopo Province, South Africa.	38
[60]	KwaZulu-Natal	A quantitative ethnobotanical survey of the Ixopo area of KwaZulu-Natal, South Africa.	9
[61]	KwaZulu-Natal	Zulu medicinal ethnobotany: new records from the Amandawe area of KwaZulu-Natal, South Africa.	16
[62]	Limpopo	An investigation into the trade of medicinal plants by muthi shops and street vendors in the Limpopo Province, South Africa.	7
[63]	KwaZulu-Natal	The medicinal ethnobotany of the Amandawe area in KwaCele, KwaZulu-Natal, South Africa.	16
[64]	Limpopo	The Ethnobotany of Central Sekhukhuneland, South Africa.	33
[65]	Limpopo	Antibacterial activities of selected medicinal plants used to treat sexually transmitted infections in Blouberg area, Limpopo Province.	11
[66]	Limpopo	Pharmacological properties of extracts from six South African medicinal plants used to treat sexually transmitted infections (STIs) and related infections.	6
[67]	Limpopo	Antimicrobial properties and phenolic contents of medicinal plants used by the Venda people for conditions related to venereal diseases.	12
[68]	Limpopo	Anti-inflammatory and mutagenic evaluation of medicinal plants used by Venda people against venereal and related infections.	12
[69]	Limpopo	Pharmacological evaluation of medicinal plants used by Venda people against venereal and related diseases.	12
[70]	Limpopo	Ethnopharmacological evaluation of a traditional herbal remedy used to treat gonorrhoea in Limpopo province, South Africa.	8
[71]	Limpopo	Plants traditionally used individually and in combination to treat sexually transmitted infections in northern Maputaland, South Africa: Antimicrobial activity and cytotoxicity.	19
[72]	Limpopo	The use of indigenous knowledge for primary health care among the Northern Sotho in the Limpopo Province.	4
[73]	Limpopo	The ethnoecological assessment of Cassia abbreviata Oliv. at Matsa village, Limpopo Province, South Africa.	1
[74]	Limpopo	Medicinal plants traded in informal herbal medicine markets of the Limpopo Province.	20
[75]	Limpopo	Antifungal activities of selected Venda medicinal plants against Candida albicans, Candida krusei and Cryptococcus neoformans isolated from South African AIDS patients.	12
[76]	Limpopo	Extraordinary Bapedi medicinal herb for gonorrhoea.	1
[77]	Limpopo	Medicinal utilisation of exotic plants by Bapedi traditional healers to treat human ailments in Limpopo Province, South Africa.	11
[78]	Limpopo	Exotic and indigenous problem plants species used by the Bapedi to treat sexually transmitted infections in Limpopo Province, South Africa.	10
[79]	Limpopo	Indigenous plant species used by Bapedi healers to treat sexually transmitted infections: Their distribution, harvesting, conservation and threats.	37
[80]	Limpopo	Herbal medicines used by Bapedi traditional healers to treat reproductive ailments in the Limpopo Province, South Africa.	21
[81]	Limpopo	Use, conservation, and present availability status of ethnomedicinal plants of Matebele-village in the Limpopo Province, South Africa.	3
[82]	Limpopo	Bapedi phytomedicine and their use in the treatment of sexually transmitted infections in Limpopo Province, South Africa.	47
[83]	Limpopo	Antibacterial activity of sixteen plant species from Phalaborwa.	2
[84]	Mpumalanga	An Exploratory study on the diverse uses and benefits of locally sourced fruit species in three villages of Mpumalanga Province, South Africa.	3
[85]	Limpopo	Ethnobotanical survey and conservation of medicinal plants used by local people Thulamela municipality, Limpopo Province.	3
[86]	Mpumalanga	Antimicrobial, antioxidant, and cytotoxicity studies of medicinal plants used in the treatment of sexually transmitted infections.	5
[87]	Mpumalanga	An Ethnobotanical Study of Medicinal Plants Used in Villages under Jongilanga Tribal council, Mpumalanga.	9
[88]	Unidentified	An antimicrobial investigation of plants used traditionally in southern Africa to treat sexually transmitted infections.	24
[89]	Unidentified	Medicinal plants of South Africa.	16
[90]	Unidentified	Medicinal plants of South Africa.	23
[91]	Unidentified	A review of Khoisan and Cape Dutch medical ethnobotany.	1
[92]	Unidentified	An ethnobotanical survey of medicinal plants in the south-eastern Karoo, South Africa.	7
[93]	Unidentified	A broad review of commercially important southern African medicinal plants.	1
[94]	Unidentified	Making the most of indigenous trees.	7
[95]	Western Cape	Medicinal plant use in the Bredasdorp/Elim region of the Southern Overberg in the Western Cape Province of South Africa.	3

**Table 2 plants-11-03241-t002:** Some South African medicinal plants and compounds with potential in treating sexually transmitted and related infections.

Plant Species	Reported Relevant In Vitro Antimicrobial Activity	Experimental Evidence Assessment	Reference(s)
*Grewia flava*	The acetone extract exhibited a minimum inhibitory concentration of 0.05 mg/mL against *Mycoplasma hominis*. Taraxerol and lupeol (Column Chromatography; NMR Mass Spectrometry and FTIR) isolated from root acetone extract exhibited MIC value of 0.03 against *Candida albicans*. Aqueous fraction from acetone roots extract exhibited MIC value of 0.04 mg/mL against *Staphylococcus aureus* isolated from wound of HIV-AIDS patient, while lupeol exhibited MIC values of 0.01 and 0.030 against *M. hominis* and *Cryptococcus neoformans*.	Positive evidence, dose dependence	[66,112,113]
*Carica papaya*	Aqueous extract from the leaves exhibited minimum inhibitory concentrations of 0.38 and 0.50 mg/mL against *Ureaplasma urealyticum* and *Oligella urealytica*, respectively, while organic extract (1:1 methanol: dichloromethane) exhibited an MIC value of 0.25 mg/mL against *Neisseria gonorhoeae.* Aqueous and ethanol extracts from flowers exhibited MIC value of 0.05 mg/mL against *Neisseria gonorrhoea.*	Positive evidence, dose dependence	[71,114]
*Helichrysum caespititium*	Organic extract from whole plant exhibited minimum inhibitory concentrations of 0.02, 0.10 and 0.06 mg/mL against *Candida albicans*, *Gardnerella vaginalis* and *Neisseria gonorrhoea*, respectively. Hexane extract from whole plant exhibited an MIC value of 0.04 mg/mL against *Neisseria gonorrhoea.*	Positive evidence, dose dependence	[115,116]
*Hypoxis haemerocallidea*	The aqueous extract from corms exhibited MIC values of 0.25 and 0.5 mg/mL against *Ureaplasma urealyticum* and *Neisseria gonorrhoea*, respectively.	Positive evidence, dose dependence	[71]
*Peltophorum africanum*	Organic extract from the roots exhibited MIC values of 0.04, 0.25 and 0.50 mg/mL against *Ureaplasma urealyticum*, *Neisseria gonorrhoea* and *Gardnerella vaginalis*.	Positive evidence, dose dependence	[71]
*Sclerocarrya birrea*	Organic extract from stem bark exhibited an MIC value of 0.25 mg /mL against both *Ureaplasma urealyticum* and *Neisseria gonorrhoea,* while aqueous extract yielded MIC values of 0.25 and 0.50 mg/mL against *Ureaplasma urealyticum* and *Neisseria gonorrhoea.*	Positive evidence, dose dependence	[71]
*Syzygium cordatum*	Aqueous extract from stem bark exhibited MIC value of 0.25 mg/mL against both *Ureaplasma urealyticum* and *Neisseria gonorrhoea*, and MIC values of 0.50 and 0.75 mg/L against *Candida albicans* and *Gardnerella vaginalis*, respectively.	Positive evidence, dose dependence	[71]
*Ximenia caffra*	Aqueous extracts from roots exhibited MIC value of 0.25 against both *Neisseria gonorhoeae* and *Ureaplasma urealyticum*, while organic extract yielded MIC values of 0.50 and 0.75 mg/mL against *Ureaplasma urealyticum* and *Oligella urealytica*, respectively.	Positive evidence, dose dependence	[71]
*Aloe ferrox*	Methanol extract from leaves and Aloin (Column Chromatography, NMR and Mass Spectrometry) exhibited MIC values of 0.5 and 0.1 mg/mL against *N. gonorhoeae*, respectively.	Positive evidence, dose dependence	[117]
*Ranunculus multifidus*	Aqueous extract exhibited MIC value of 0.02 mg/mL against *Ureaplasma urealyticum.*	Positive evidence, dose dependence	[71]
*Withania somnifera*	Methanol root extract exhibited MIC value of 0.5 mg/mL against *N. gonorrhoeae*.	Positive evidence, dose dependence	[117]
*Bowiea volubilis*	The ethanol extract from bulbs exhibited MIC value of 3.125 mg/mL against *Staphylococcus aureus.*	Inconclusive evidence, dose dependence	[39,40,89]

**Table 3 plants-11-03241-t003:** Possible mixtures used in the treatment of sexually transmitted and related infections.

Plant Species	Mixture or Possible Quantities	Therapeutic Indication	Reference
*Terminalia sericea*, *Cassine transvaalensis*, *Elephantorrhiza burkei*, *Rauvolvia caffra* and *Andredera cordifolia*	Equal quantities of the plants	Unidentified STIs	[17]
*Bauhinia galpinii*, *Elephantorrhiza burkei* and *Cassia abbreviata*	Equal quantities of the roots are boiled and taken orally	Impotence	[19]
*Peltophorum africanum, Elephantorrhiza burkei, Cassia abbreviata,* and three nodes of *Cissus quadrangulari*	Equal quantities of the plant species and three nodes of *Cissus quadrangularis*	Dropsy/syphilis	[19]
*Cassia Abbreviata, Jatropha zeyheri, Cissus quadrangularis, Elephantorrhiza burkei, Peltophorum africanum, Waltheria indida, Urginea sanguinea* and *Harpagophythum procumbens*	Equal quantities of the plant species and three nodes of *Cissus quadrangularis*	Dropsy/syphilis	[65]
*Anredera cordifolia*, *Catharanthus roseus*, *Datura stramonium*, *Gomphocarpus fruticosus, Prunus persica*, *Senna italica*, *Senna petersiana*, and *Solanum panduriforme*	Different plant parts	Gonorrhoea and opportunistic infections associated with HIV-AIDS	[70]

## Data Availability

Data for this article can be found online Appendix A.

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
