# Peer review of "An Inventory of South African Medicinal Plants Used in the Management of Sexually Transmitted and Related Opportunistic Infections: An Appraisal and Some Scientific Evidence (1990–2020)"

_plants, 2022, doi:10.3390/plants11233241_

Round 1
Reviewer 1 Report
Feedback on the review «An Inventory of South African Medicinal Plants Used in The Management of Sexually Transmitted and Related Opportunistic Infections: An Appraisal and Some Scientific Evidence (1990- 2020)».
The review is devoted to the analysis of the literature on South African medicinal plants that can be used to treat sexually transmitted infections.The review consists of the following sections: Introduction, which discusses the historical aspect, raises the problem of using African plant raw materials as a possible alternative to traditional medicines, as well as the use of plant raw materials as an immunity stimulant for the prevention of STI. The Results and Discussions section summarizes the results taken from the literature. This is the main part of the review, including 6 figures and 2 tables, where the families and plant species most often used for the treatment of STIs and related infections are considered in some detail. The section is pretty detailed. From minor remarks: in my opinion, Figure 6 is better to modify by adding the composition of the ingredients to each agent used.
The review ends with a conclusion showing that the highest citations of plants from various literature data for the treatment of STIs and related infections are Catharanthus roseus and Peltophorum africanum.
However, we still would like to clarify the position of the authors themselves regarding the use of plant extracts for the treatment of STIs and related infections.Do the authors believe that plant extracts can currently replace the drugs used to treat STIs and related infections, or can they be used only for their prevention?In my opinion, the review should be based around the position of the authors themselves.More material is also needed on clinical studies and comparison of the effects of plant extracts with currently used traditional drugs.
In general, the work looks quite original.If the authors clarify their position and respond to the aforementioned comments, the review may be published in the “Plants” journal.
Author Response
Good day
Please find the attached

Reviewer 2 Report
Congratulations!
Author Response
Good day
The authors are thankful that you are satisfied with the paper submitted to your esteemed Journal, Plants
Reviewer 3 Report
The manuscropt has been submitted to the Section Plant Systematics, Taxonomy, Nomenclature and Classification, however it deals with none of these topics. In its present form the MS is of limited scientific interest and in my opinion is better suited for submission to journals whose main topic is herbal medicines.
Author Response
No comments requested